# The Effect of Social Media Activities on Brand Loyalty for Banks: The Role of Brand Trust

Sulaiman Althuwaini

Business Studies College, Arab Open University-KSA, Hittin, Riyadh 11681, Saudi Arabia;
s.althuwaini@arabou.edu.sa

**Abstract:** The aim of this research was to examine the impact of social media marketing activities on brand trust and brand loyalty in the banking sector. Based on an online survey of 252 users who follow banking services suppliers on social media located in Saudi Arabia, data were gleaned and analyzed via Smart-PLS (3.0). The findings showed that social media marketing activities, namely customization, entertainment, and promotions, were reported to have the highest impact on trust and loyalty. This study emphasizes the role of trust as mediating brand loyalty in social media marketing. The implications for marketing managerial and future research are discussed.

**Keywords:** social media marketing; brand trust; brand loyalty

## 1. Introduction

Social media marketing (SMM) is the practice of using social media to attain organizational goals for businesses (Felix et al. 2017, p. 123). SMM has been investigated using many methods and from many perspective by various researchers (Kim and Ko 2012; Seo and Park 2018). Customers are more drawn to brands that produce enjoyable consumer and service experiences. Marketers have selected social media as a crucial channel for reaching their target market. Users of social media may connect, communicate, and have discussions with people there. This indicates that businesses on social media are in a unique position to offer engaging and heartfelt experiences. In the banking industry, social media has become a preferred interactive choice to provide customers contact and communication. Banks can get closer to customers by developing a presence on social media. In addition, social and digital innovations allow banks to reach their actual and prospective customers with unique and memorable brand experiences (Khan et al. 2016).

As banks have attempted to integrate new innovations channels, there has also been a gradual shift in informational and transactional touchpoints between banks and their customers. Social media enables businesses to contact and communicate with customers in a more cost-effective and efficient way than traditional channels (Khanum et al. 2016). It provides multiple opportunities for customer interaction through likes, comments, shares, and views of products (Ahmed et al. 2018). When companies connect with customers through social media, they can achieve customer retention and loyalty (Kim and Ko 2012). Companies can interact with customers at any time, and also user-generated content can be facilitated. Therefore, social media marketing supports companies to achieve marketing strategies to sustain businesses. According to Laroche et al. (2012), when a brand implements effective marketing efforts and relevant content, it will positively affect customer loyalty.

Despite increased scholarly interest, quantifying the influence of social media marketing activities on a brand's success remains an ongoing issue, despite the fact that social media offers new opportunities and advantages for brand management (Kaplan and Haenlein 2010; Habibi et al. 2014). Furthermore, research that analyzes the branding literature's perspective on social media marketing's effects is still primarily exploratory and devoid of empirical data (Hollebeek et al. 2014). Accordingly, this study's objective is to provide

a better understanding by investigating the impact of social media marketing activities (entertainment, interaction, trendiness, customization, word of mouth and promotion) on the brand, in particular brand loyalty. The main research question is:

-    What is the impact of the elements of social media marketing activities that influence customers' brand loyalty?

The structure of this study is as follows: the literature review discusses the main concepts, followed by the presentation of the research model and hypotheses. The next part of the study describes the research methodology, and then data analysis and results are outlined. After that, the discussion part is presented. In the final part of this study, theoretical and managerial implications as well as research limitations and future research directions are discussed.

## 2. Literature Review

### 2.1. Social Media Marketing

Social media is represented by online platforms and groups that promote social interaction by allowing users to share their thoughts, experiences, and observations (Alalwan et al. 2017). Social media provides tremendous potential to engage buyers individually, as well providing as a way to cement the customer–brand relationship (Ismail et al. 2018). It provides frequent communication with other users to disperse information and creates awareness among people. Users of social media can readily share their brand experiences and exposure and exchange information with one another, influencing brand preferences and boosting brand information among other community members (Laroche et al. 2012). According to Lim et al. (2020), social media is an interaction medium that allows for the development of trust, which is crucial for the establishment of brand equity. Social media networks involve users using social media with the purpose of connecting with people (Ebrahim 2019).

A considerable amount of research has been published on social media's benefits. For example, Laroche et al. (2012) expressed that social media facilitates information sharing and enhances customers' bonds with each other, and it strengthens customers' relationships with the brand, the product, the company and other customers. Through a large-scale social network, social media provides a unique chance for word-of-mouth marketing usage to a massive audience, boosting consumer-to-consumer communications and expanding brand awareness (Kozinets et al. 2010). Social media and businesses unlock tremendous potential by personalizing brands, increasing capital and influencing the buyer's willingness to buy (Lim et al. 2020).

Many business sectors keep in continual contact with their customers by using social media for marketing and promotion. It is critical to establish and apply marketing activities in social media to obtain efficient and functional value in a company's marketing activities (Bilgin 2018). Previous studies have indicated different attempts to conceptualize social media marketing activities in different business sectors (Sharawneh 2020; Bilgin 2018; Ibrahim and Aljarah 2018; Seo and Park 2018; Yadav and Rahman 2017; Kim and Ko 2012). Kim and Ko (2012) identified five dimensions pertaining to social media marketing activities (SMMA); trendiness, entertainment, interaction, word of mouth and customization. These factors were studied in a luxury fashion brand. Seo and Park (2018) conducted a study within the airline service industry by investigating entertainment, interaction, trendiness, customization and perceived risk as social media marketing activities. Yadav and Rahman (2017) developed and validated five perceived SMMA scales for interactivity, informativeness, personalization, trendiness, and word of mouth. Other researchers have also examined social media marketing activities differently depending on the purpose of their study (Bilgin 2018; Ismail et al. 2018).

Social media is considered a valuable source for providing entertainment to consumers. Entertainment represents the fun and pleasure derived from using social media (Alalwan et al. 2017). The dimension of interaction describes how users contribute to brands on social media platforms. Trendiness refers to a brand the capability of collaborating and

providing new and up-to-date topics on social media (Naaman et al. 2011). Kim and Ko (2012) showed that "newest information" and "trendy" are two important phrases that reflect trendiness. Customization as a component refers to the act of cultivating customer satisfaction based on the contact between the business and individual users (Seo and Park 2018). Word of mouth is the exchange of product- and brand-related marketing messages and meanings (Loureiro et al. 2017). This involves exchanging opinions about a brand with other users and disseminating brand information through the publishing of photos, videos, or stories. Users tend to consider purchasing a brand according to recommendations or experiences from followers on social media. The advertising as a component refers to advertising and promotional campaigns that businesses have implemented through social media to increase sales and develop the customer portfolio. Promotional campaigns and online advertisements were reported as crucial components of SMM activities (Bilgin 2018; Alalwan et al. 2017).

## 2.2. Brand Trust

Trust has been viewed as a fundamental variable in the exchange network between a company and its customers. Brand trust refers to the state where consumers are willing to rely on the brand because of its reliability and integrity to perform its stated function (Chaudhuri and Holbrook 2001). It denotes confidence that the relational participant in a trade will not take advantage of another's vulnerabilities. As a result, instinctively trusting a brand implies a high likelihood or expectation that the brand would result in favorable outcomes for the consumer. There are a number of studies which addressed the role of trust in the online environment (Laroche et al. 2012). Trust acts as a mechanism for reducing consumers' perceived risk in e-commerce. It reduces the perceived risk of facing a negative outcome of an interaction by lowering information complexity (Mayer et al. 1995).

In a study conducted by Habibi et al. (2014) where they investigated how brand community on social media can influence brand trust, the findings showed that three brand community relationships (i.e., customer–brand, customer–product, and customer–company) positively influence brand trust. However, the findings also showed that the relationship between customers and other customers negatively influences brand trust. Different consumers join social media brand communities to gain vital knowledge or to better use the brand's products or services, regardless of whether they have prior experience with the brand. Consumers who have had positive experiences with the brand and are familiar with it are more likely have positive attitudes toward the brand's honesty and responsibility, which can positively affect their perception of the brand and facilitate the process of forming associations with it (Habibi et al. 2014).

## 2.3. Brand Loyalty

The benefits of loyal customers are well documented in the marketing literature. Loyal customers tend to make a greater volume of purchases on a more frequent basis, are less prone to defecting to a competitor, and through word-of-mouth communication will recruit more customers for the organization (Yoo and Bai 2013). While no formal definition for customer loyalty has been established, it appears that most scholars agree that the construct of customer loyalty includes both a behavioral and an attitude dimension. Behavioral loyalty relates to consumers' repurchasing behavior or intent regarding a specific brand, while attitudinal loyalty relates consumers' perception of specific products or services. Focusing on these two perspectives, brand loyalty is defined in this study as the deeply held commitment toward rebuying the brand in the future, regardless of situational factors (Chaudhuri and Holbrook 2001). Based upon this definition, behavioral loyalty tends to lead to a high market share, while attitudinal loyalty leads to higher relative brand value (Taylor et al. 2004).

In online research, several antecedents of customer loyalty have already been identified (Ariffin et al. 2021; Nguyen and Khoa 2019; Yee and Faziharudean 2010). Due to the characteristics of the online environment, companies need to focus on factors allied with the

online setting, such as interactivity, convenience, relevance, perceived value and customization (Zhu and Chang 2016; Jahn and Kunz 2012). Previous studies have confirmed the relationship between the positive influence of social media marketing activities and brand loyalty (Ibrahim 2021; Ebrahim 2019). Such online activities are brand stimuli which affect the experiences of consumers and strengthen their relationship with the service providers, which as a result improve their behavioral responses, which can be represented by loyalty and satisfaction (Laroche et al. 2012). This pertains to the swiftness of communication and information that is exchanged via social media. Social networks are an attractive marketing tool for companies to provide a new landscape for increasing relationships with customers (Ebrahim 2019). Based on the previous discussion, this study proposed the following hypotheses and Figure 1 shows the research model:

**H1:** *Social media marketing activities have a significant positive effect on brand loyalty.*

**H2:** *Social media marketing activities have a significant positive effect on brand trust.*

**H3:** *Brand trust has a significant positive effect on brand loyalty.*

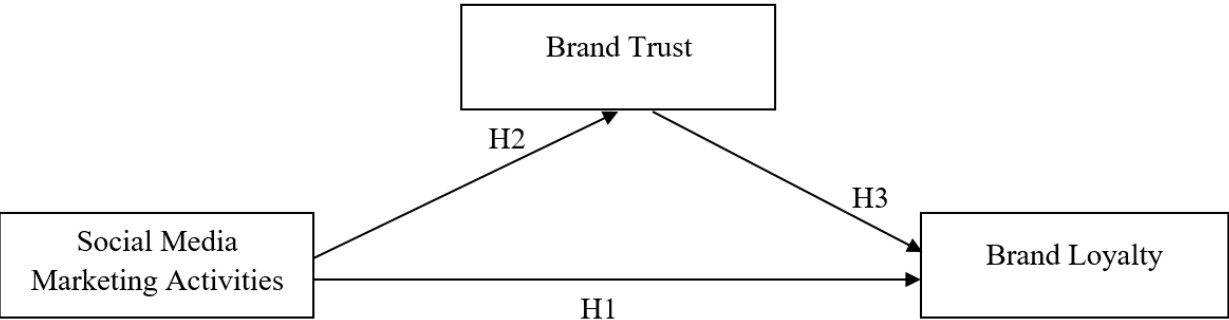

**Figure 1.** Research model.

### 3. Methodology

This study targeted social media users and it applied a quantitative approach. A survey was developed based on previous studies on social media marketing (Bilgin 2018; Kaur et al. 2018; Seo and Park 2018; Yadav and Rahman 2017; Kim and Ko 2012; Yoo et al. 2000). Brand trust items were gathered from (Chaudhuri and Holbrook 2001), and brand loyalty items were developed in accordance with (Ebrahim 2019; Yoo et al. 2000; Aaker 1991). Measurements of variables used a 5-point Likert-scale, from 1 'strongly disagree' to 5 'strongly agree'.

The primary data were collected using a self-administrated online questionnaire, and all the items were developed and adapted from previous studies. The questionnaire went through a revision by a panel of researchers during the pilot study. The aim was to check the validity of the measurement items. The questionnaire consisted of three main parts; the first contained a brief introduction of the questionnaire's purpose, the second was related to the respondents' demographic profile, and the third was divided into three subsections; the first was related to measuring social media activities, the second to measuring brand trust and the third to measuring brand loyalty. The questionnaire was published online using Google Forms. The setting was adjusted to allow respondents to answer all the questions before going onto the next part. The study targeted commercial Saudi Arabian banks' individual customers in a social media context over the period of three months from April 2022 to June 2022. The link to the survey was distributed among different users of social media who followed their banking service providers. Respondents were asked to add their bank's name in the first part of the questionnaire.

The population of this study included social media users. Considering information limitations regarding users in the population, the convenience sampling method was applied, which is a sampling method that is not based on probability. A total of 283

questionnaires were received, 252 of which were complete and valid for statistical analysis. Table 1 shows the demographic characteristics of the respondents.

**Table 1.** Demographic profile.

| Characteristics | N | % |
|---|---|---|
| Gender | | |
| Male | 88 | 34.9 |
| Female | 164 | 65.1 |
| Age Group (yr.) | | |
| 18–20 | 78 | 31 |
| 21–30 | 130 | 51.6 |
| 31–40 | 40 | 15.9 |
| 41–50 | 2 | 0.8 |
| 51–60 | 2 | 0.8 |
| Job Status | | |
| Employed | 56 | 22.2 |
| Freelancer | 14 | 5.6 |
| Student | 176 | 69.8 |
| Others | 6 | 2.4 |
| Education Status | | |
| Bachelor level | 164 | 65.1 |
| Master level | 4 | 1.6 |
| PhD level | 2 | 0.8 |
| Others | 82 | 32.5 |
| Twitter Usage | | |
| Daily | 156 | 61.9 |
| Weekly | 32 | 12.7 |
| Monthly | 4 | 1.6 |
| In need | 60 | 23.8 |
| Twitter Experience | | |
| Less than a year | 34 | 13.5 |
| 2–5 year | 90 | 35.7 |
| 6–10 year | 90 | 35.7 |
| +10 year | 38 | 15.1 |

### 3.1. Data Analysis

Following Hair et al. (2019), a two-step procedure was implemented to run Smart-PLS analysis 3.0 developed by Ringle et al. (2005), a measurement model and a structural model.

### 3.2. Measurement Model

Hair et al. (2019), and Herjanto et al. (2021) proposed testing internal consistency reliability, convergent validity, and discriminant validity in the measurement model analysis. The Cronbach's alpha was calculated to measure internal consistency reliability, and a value with a range of 0.716 to 0.923 (see Table 2) was obtained. Standardized factor loadings, composite reliability, and the average variance extracted (AVE) were calculated (Anderson and Gerbing 1988). Factor loading for each construct ranged from 0.644 to 0.926. The com-

posite reliability (CR) ranged from 0.824 to 0.946, and the average variance extracted (AVE) ranged from 0.542 to 0.856, and thus was above the recommended level of 0.50 (Anderson and Gerbing 1988; Hair et al. 2019). The Fornell–Larcker procedure (Fornell and Larcker 1981) and the heterotrait–monotrait (HTMT) method (Henseller et al. 2016) were followed to test discriminant validity (see Table 3). The results of the Fornell–Larcker calculation justified that the square root of AVE between each set of the construct was higher than the association estimated between constructs, thus demonstrating that discriminant validity was recognized (Fornell and Larcker 1981). The heterotrait–monotrait (HTMT) ratio of associations clarified that all values of HTMT were lower than the proposed level of 0.85, thus allowing us to declare acceptable discriminant validity for all constructs (Hair et al. 2019).

**Table 2.** Construct validity.

| Constructs | Items | Factor Loadings | $\alpha$ | CR | AVE |
|---|---|---|---|---|---|
| Customization (CUS) | My bank's social media offer a customized information search | 0.914 | 0.923 | 0.946 | 0.813 |
| | My bank's social media provide customized service | 0.893 | | | |
| | The social media of my bank provided the information that I needed | 0.920 | | | |
| | The information that I need can be found in the social media of my bank | 0.880 | | | |
| Entertainment (ENT) | Following my bank's social media is fun | 0.807 | 0.863 | 0.907 | 0.708 |
| | Content of my bank's social media seems interesting | 0.840 | | | |
| | Following my bank's social media brings me hopefulness | 0.888 | | | |
| | Following my bank's social media brings me happiness | 0.830 | | | |
| | My bank's social media enables information-sharing with others | 0.804 | 0.764 | 0.864 | 0.679 |
| Interaction (INT) | Conversation or opinion exchange with others is possible through my bank's social media | 0.857 | | | |
| | It is easy to provide my opinion through my bank's social media | 0.812 | | | |
| | I follow my bank's social media for promotional campaigns | 0.838 | 0.846 | 0.896 | 0.684 |
| Promotion (PRO) | I like social media promotions published by my bank's social media | 0.811 | | | |
| | Promotional information on my bank's social media is useful | 0.852 | | | |
| | My bank frequently offers price discounts | 0.807 | | | |
| | Content of my bank's social media is the newest information | 0.926 | 0.832 | 0.923 | 0.856 |
| | Using my bank's social media is very trendy | 0.925 | | | |
| | I would like to upload content from my bank's social media on my blog | 0.674 | 0.716 | 0.824 | 0.542 |
| Word of Mouth (WOM) | I am going to spread positive WOM about my bank's social media | 0.825 | | | |
| | I will recommend my bank to other customers | 0.788 | | | |
| | I will point out the positive aspects of my bank if anybody criticizes it | 0.644 | | | |
| Trust | My bank is honest | 0.831 | 0.856 | 0.903 | 0.699 |
| | My Bank's promises are real | 0.876 | | | |
| | My bank works for my happiness | 0.785 | | | |
| | My bank works hard to satisfy me | 0.851 | | | |
| Loyalty | I will suggest my bank to other consumers | 0.837 | 0.914 | 0.934 | 0.702 |
| | I would love to recommend my bank to my friends | 0.879 | | | |
| | I consider myself to be loyal to my bank | 0.858 | | | |
| | My bank would be my first choice | 0.865 | | | |
| | I intend to keep purchasing the services offered by my bank | 0.732 | | | |
| | I am loyal to my bank | 0.846 | | | |

**Table 3.** Discriminant validity.

| Construct | CUS | ETN | INT | Loyalty | PRO | TRE | Trust | WOM |
|---|---|---|---|---|---|---|---|---|
| Fornell-Larcker Criterion | | | | | | | | |
| CUS | 0.902 | | | | | | | |
| ETN | 0.621 | 0.842 | | | | | | |
| INT | 0.671 | 0.613 | 0.824 | | | | | |
| Loyalty | 0.417 | 0.411 | 0.327 | 0.838 | | | | |
| PRO | 0.561 | 0.598 | 0.568 | 0.338 | 0.827 | | | |
| TRE | 0.699 | 0.544 | 0.688 | 0.468 | 0.572 | 0.925 | | |
| Trust | 0.406 | 0.399 | 0.298 | 0.776 | 0.337 | 0.400 | 0.836 | |
| WOM | 0.552 | 0.665 | 0.537 | 0.582 | 0.545 | 0.585 | 0.517 | 0.736 |
| Heterotrait-Monotrait Ratio (HTMT) | | | | | | | | |
| CUS | | | | | | | | |
| ETN | 0.694 | | | | | | | |
| INT | 0.792 | 0.751 | | | | | | |
| Loyalty | 0.454 | 0.468 | 0.384 | | | | | |
| PRO | 0.633 | 0.696 | 0.706 | 0.393 | | | | |
| TRE | 0.797 | 0.638 | 0.760 | 0.535 | 0.681 | | | |
| Trust | 0.452 | 0.464 | 0.363 | 0.769 | 0.395 | 0.474 | | |
| WOM | 0.660 | 0.730 | 0.712 | 0.715 | 0.691 | 0.749 | 0.656 | |

### 3.3. Structural Model

Following Hair et al.'s (2019) recommendations, three methods for reporting structural model were calculated: path coefficient ($\beta$), coefficient of determination ($R^2$), and effect size ($f^2$). Table 4 shows the hypotheses testing. Social media marketing activities have a significant relationship with trust and loyalty, and thus H1 and H2 were accepted ($\beta = 0.490$; $\beta = 0.188$). Trust has a significant relationship with a loyalty, and thus H3 was accepted ($\beta = 0.684$). For the coefficient of determination, loyalty was explained at a rate of 63% by trust ($R^2 = 0.629$) and social media marketing activities ($R^2 = 0.240$). The size effect ($f^2$) values demonstrate that social media marketing activities have a high effect on trust (0.316) and a low effect on loyalty (0.072), while trust has a high effect on loyalty (0.400).

**Table 4.** Structural model.

| Hypotheses | Beta | t-Statistics | *p*-Values | $R^2$ | $f^2$ | Supported |
|---|---|---|---|---|---|---|
| H1. Social Media Marketing Activities $\rightarrow$ Trust | 0.490 | 6.860 | 0.000 | 0.629 | 0.316 | Yes |
| H2. Social Media Marketing Activities $\rightarrow$ Loyalty | 0.188 | 3.759 | 0.000 | 0.240 | 0.072 | Yes |
| H3. Trust $\rightarrow$ Loyalty | 0.684 | 13.973 | 0.000 | 0.240 | 0.400 | Yes |

### 4. Discussion

The results of this study shed light on social media marketing activities' impact on enhancing important branding objectives including brand trust and brand loyalty, adding to the expanding body of literature in the field. It remains difficult to measure and conceptualize SMM activities (Yadav and Rahman 2017). Several studies in the earlier literature verified the characteristics of social media marketing activities produced by Kim and Ko (2012). The activities that drive consumers to use social media are defined by the overall influence of these factors. In this study, findings showed that social media marketing activities have a positive effect on brand trust and brand loyalty. Customization was proved to have the highest impact on trust and loyalty (0.278), followed by entertainment (0.231), promotions (0.208), WOM (0.200), and interaction (0.156), while trendiness had the lowest impact (0.149). From the consumer point of view, users engage in online platforms in order to obtain tailored services and entertaining content while they share their experiences and recommendations with others. This supports previous studies such as Lim et al. (2020) and Ebrahim (2019). For a positive impact on the brand, the communication efforts must be interactive, customized and entertaining. This model also confirms that SMM activities

enrich brand trust and have an important role in producing brand loyalty. Consumers who trust the brand's SMM activities the most are more likely to recommend the brand. This is an important contribution of this study, as loyal customers hold revisiting intentions toward the bank's social media platform, triggered by updated SMM activities.

### 4.1. Theoretical and Managerial Implications

This study attempts to contribute to the current literature on social media marketing and brand loyalty and has distinct theoretical implications. Initially, a unique model which comprises six features of social media marketing activities was developed to verify their impacts on brand loyalty. By looking at the extant literature, it can be noticed that these dimensions were used separately in different studies. This paper measures their impact on brand loyalty collectively and aims to provide better insights for Middle East region, as previous studies on this particular topic are scarce. In theory, the findings are consistent with the social media literature in that social media marketing activities have a significant relationship with trust and loyalty.

The conclusions have important marketing implications for those working in the banking sector. The findings demonstrate that marketing professionals in the banking sector brands should use social media marketing activities to sustain loyalty. The results particularly support the significance of using interactive and informative social media content to increase consumers' loyalty to banking brands. In this context, banks should pay attention to the design of social media activities that support customer engagement, ensuring they are original and creative.

### 4.2. Limitation and Future Research Directions

This paper has some limitations that can be taken into account in subsequent investigations. First, this study used a quantitative approach to gather consumer data from Saudi Arabia, which may limit the generalizability of the findings. Future research can take into consideration a longitudinal design to get around this limitation and collect information for the chosen variables from consumers across various cultural backgrounds. This study was conducted in Saudi Arabia, and so it is suggested for researchers to perform a comparative study in other regions. Second, the research domain was in the banking industry; consequently, future studies can focus on other industry domains to confirm these findings. Finally, six components of social media marketing activities were tested to determine their impacts on brand loyalty. Consequently, upcoming studies can examine other elements, such as mediator variables (i.e., customer satisfaction) or moderator variables (i.e., customer characteristics).

**Funding:** The researcher would thank Arab Open University-KSA for funding this research. Fund number RGP-22-01.

**Institutional Review Board Statement:** This research was funded by [Arab Open University-KSA] grant number [RGP-22-01].

**Informed Consent Statement:** Informed consent was obtained from all subjects involved in the study.

**Data Availability Statement:** The data presented in this study are available on request from the corresponding author. The data are not publicly available due to the personal nature of the thesis dataset and the research journal.

**Conflicts of Interest:** The author declares no conflict of interest.

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
