# Peer review of "The Effect of Social Media Activities on Brand Loyalty for Banks: The Role of Brand Trust"

_admsci, doi:10.3390/admsci12040148_

Round 1

Reviewer 1 Report

manuscript is clear, relevant for the field ,  cited references mostly recent publications, data interpreted appropriately and consistently throughout the manuscript, conclusions consistent with the evidence and arguments presented

Author Response

Dear Reviewer 1

First, I would like to thank you for your valuable feedback which improved my research paper. I have done your suggestions and please can you look it again and approve it. I am happy for your constructive feedback. 

Reviewer 2 Report

Dear Author(s),

Please find below my concerns and recommendations regarding your manuscript proposal entitled "The Effect of Social Media Activities on Brand Loyalty for Banks: The Role of Brand Trust".

1. The Introduction section is pretty-well written, but you need to also include the following important aspects: the research goal and the research question(s).

I also recommend you to shortly describe the structure of the article at the end of the Introduction chapter.

2. In the section "3. Methodology" at rows 165-168 you say: "The study targeted commercial banks’ customers on social media context, over the period of three months from April 2022 to June 2022. The link of the survey was distributed among different users of social media. The population of this research is considered of social media users on Twitter."

You tried to describe the sample of your research, but there are some unclear aspects:

- the commercial bank's customers are individual or/and companies?

- which banks were involved in your study? What country/region?

- how did you access the customers via Twitter? How did you contact them? Did the bank(s) tell you who are their customers?!?

Please clarify these aspects in your manuscript proposal.

3. I recommend you to include an appendix in your paper where you present the items and the constructs. For example, the items TRU1, TRU2, TRU3, TRU4 appear in the table 2, but the readers don't know their meaning.

This recommendations is for all the items used in your research.

4. The section between rows 259-305 must be edited according to the journal's requests. At this moment, this sequence is just a standard template, but it must be completed with real data.

5. In "Table 4. Structural model" I recommend you to add a new distinct column entitled "Supported" and complete the values with "Yes" or "No" for each hypothesis (in this case "Yes"), so that the readers see the status of every research hypothesis.

Dear Author(s),

Please consider all the above remarks as being constructive recommendations in order to improve the general quality of your manuscript proposal.

Kind Regards!

Author Response

Dear Reviewer 2

Hope you are fine,

I would like to thank you for your valuable feedback which improved my research paper. I have done your suggestions and please can you look it again and approve it.

Please note that I could not reach banks' customers' via Tiwitter and I did consult other researchers who guided me to share my questionnaire link among social media. In the survey, respondents have a question regarding if they are following their banks' accounts on Twitter. If yes, they can proceed to answer the questionnaire. I will be happy if you can accept my corrections and approve my my paper. 

Reviewer 3 Report

The paper has rather very general or simplistic research model.

Items in model constructs should be explained in more details in the paper.

Research sample contains mainly female students, so it might affect the results, but their higher presence (above average) in the social media is obvious.

One would expect that hypotheses would be presented in the research section of the paper.

Reseachers use standard methods in this paper.

Line 212 is rather empty for no obvious reason.

DOI identification for all references should be provided where it is available.

English in the paper is good, but few mistakes can be found (e.g. line 141 "and to Figure 1. shows" - "to" is redundant  word).

Author Response

Dear Reviewer 3

Hope you are fine

First, I would like to thank you for your valuable feedback which improved my research paper. I have done your suggestions and please can you look it again and approve it. I am happy for your constructive feedback. 

Round 2

Reviewer 2 Report

Dear Author(s),

I have read the new version of the manuscript proposal and I appreciate your effort to improve the article.

I consider you addressed most of my constructive recommendations from the previous round of review and the new version of the proposal is improved.

Now I have only one minor remark regarding the future research directions. I recommend you to also include as a further direction the idea of implementing a comparative study between different regions/countries.

Kind Regards!

Author Response

Dear Reviewer 2

Thank you for your feedback, I have worked on your suggestions.

Kind Regards,

Reviewer 3 Report

Changes in the paper fully answered my comments / remarks to the original manuscript.

Author Response

Dear Reviewer 3

Thank you for your efforts, and for your constructive feedback.

Kind Regards,